# Discovering Hierarchical Achievements in Reinforcement Learning via Contrastive Learning

**Seungyong Moon**[1], **Junyoung Yeom**[1], **Bumsoo Park**[2], **Hyun Oh Song**[1]*
[1]Seoul National University, [2]KRAFTON
`{symoon11,yeomjy,hyunoh}@mllab.snu.ac.kr`
`bumsoo.park96@krafton.com`

## Abstract

Discovering achievements with a hierarchical structure in procedurally generated environments presents a significant challenge. This requires an agent to possess a broad range of abilities, including generalization and long-term reasoning. Many prior methods have been built upon model-based or hierarchical approaches, with the belief that an explicit module for long-term planning would be advantageous for learning hierarchical dependencies. However, these methods demand an excessive number of environment interactions or large model sizes, limiting their practicality. In this work, we demonstrate that proximal policy optimization (PPO), a simple yet versatile model-free algorithm, outperforms previous methods when optimized with recent implementation practices. Moreover, we find that the PPO agent can predict the next achievement to be unlocked to some extent, albeit with limited confidence. Based on this observation, we introduce a novel contrastive learning method, called *achievement distillation*, which strengthens the agent's ability to predict the next achievement. Our method exhibits a strong capacity for discovering hierarchical achievements and shows state-of-the-art performance on the challenging Crafter environment in a sample-efficient manner while utilizing fewer model parameters.

## 1 Introduction

Deep reinforcement learning (RL) has recently achieved remarkable successes in solving challenging decision-making problems, including video games, board games, and robotic controls [31, 42, 18, 36]. However, these advancements are often restricted to a single deterministic environment with a narrow set of tasks. To successfully deploy RL agents in real-world scenarios, which are constantly changing and open-ended, they should generalize well to new unseen situations and acquire reusable skills for solving increasingly complex tasks via long-term reasoning. Unfortunately, many existing algorithms exhibit limitations in learning these abilities and tend to memorize action sequences rather than truly understand the underlying structures of the environments [28, 24].

To assess the abilities of agents in generalization and long-term reasoning, we focus on the problem of discovering hierarchical achievements in procedurally generated environments with high-dimensional image observations. In each episode, an agent navigates a previously unseen environment and receives a sparse reward upon accomplishing a novel subtask labeled as an *achievement*. Importantly, each achievement is semantically meaningful and can be reused to complete more complex achievements. Such a setting inherently demands strong generalization and long-term reasoning from the agent.

Previous work on this problem has mainly relied on model-based or hierarchical approaches, which involve explicit modules for long-term planning. Model-based methods employ a latent world model that predicts future states and rewards for learning long-term dependencies [20, 21, 1, 45]. While

---

*Corresponding author

these methods have shown effectiveness in discovering hierarchical achievements, particularly in procedurally generated environments, they are constructed with large model sizes and often require substantial exploratory data, which limits their practicality. Hierarchical methods aim to reconstruct the dependencies between achievements as a graph and employ a high-level planner on the graph to direct a low-level controller toward the next achievement to be unlocked [43, 10, 49]. However, these methods rely on prior knowledge of achievements (*e.g.*, the number of achievements), which is impractical in open-world scenarios where the exact number of achievements cannot be predetermined. Additionally, they necessitate a significant number of offline expert data to reconstruct the graph.

To address these issues, we begin by exploring the potential of proximal policy optimization (PPO), a simple and flexible model-free algorithm, in discovering hierarchical achievements [39]. Surprisingly, PPO outperforms previous model-based and hierarchical methods by adopting recent implementing practices. Furthermore, upon analyzing the latent representations of the PPO agent, we observe that it has a certain degree of predictive ability regarding the next achievement, albeit with high uncertainty.

Based on this observation, we propose a novel self-supervised learning method alongside RL training, named *achievement distillation*. Our method periodically distills relevant information on achievements from episodes collected during policy updates to the encoder via contrastive learning [34]. Specifically, we maximize the similarity in the latent space between state-action pairs and the corresponding next achievements within a single episode. Additionally, by leveraging the uniform achievement structure across all environments, we maximize the similarity in the latent space between achievements from two different episodes, matching them using optimal transport [5]. This learning can be seamlessly integrated into PPO by introducing an auxiliary training phase. Our method demonstrates state-of-the-art performance in discovering hierarchical achievements on the challenging Crafter benchmark, unlocking all 22 achievements with a budget of 1M environment steps while utilizing only 4% of the model parameters compared to the previous state-of-the-art method [19].

## 2 Preliminaries

### 2.1 Markov decision processes with hierarchical achievements

We formalize the problem using Markov decision processes (MDPs) with hierarchical achievements [49]. Let $\mathbb{M}$ represent a collection of such MDPs. Each environment $\mathcal{M}_i \in \mathbb{M}$ is defined by a tuple $(\mathcal{S}_i, \mathcal{A}, \mathcal{G}, p, r, \rho_i, \gamma)$. Here, $\mathcal{S}_i \subset \mathcal{S}$ is the image observation space, which has visual variations across different environments, $\mathcal{A}$ is the action space, $\mathcal{G}$ is the achievement graph with a hierarchical structure, $p : \mathcal{S} \times \mathcal{A} \to \mathcal{P}(\mathcal{S})$ is the transition probability function, $r : \mathcal{S} \times \mathcal{A} \times \mathcal{S} \to \mathbb{R}$ is the achievement reward function, $\rho_i \in \mathcal{P}(\mathcal{S}_i)$ is the initial state distribution, and $\gamma \in [0, 1]$ is the discount factor.

The achievement graph $\mathcal{G} = (\mathcal{V}, \mathcal{E})$ is a directed acyclic graph, where each vertex $v \in \mathcal{V}$ represents an achievement and each edge $(u, v) \in \mathcal{E}$ indicates that achievement $v$ has a dependency on achievement $u$. To unlock achievement $v$, all of its ancestors (*i.e.*, achievements in the path from the root to $v$) must also be unlocked. When an agent unlocks a new achievement, it receives an achievement reward of 1. Note that each achievement can be accomplished multiple times within a single episode, but the agent will only receive a reward when unlocking it for the first time. Specifically, let $b \in \{0, 1\}^{|\mathcal{V}|}$ be a binary vector indicating which achievements have been unlocked and $c : \mathcal{S} \times \mathcal{A} \times \mathcal{S} \to \mathcal{V} \cup \{\emptyset\}$ be a function determining whether a transition tuple results in the completion of an achievement. Then, the achievement reward function is defined as

$$r(s_t, a_t, s_{t+1}) = \begin{cases} 1 & \text{if } \exists v_i \in \mathcal{V} : b[i] = 0, c(s_t, a_t, s_{t+1}) = v_i \\ 0 & \text{otherwise.} \end{cases}$$

This reward structure provides an incentive for the agent to explore the environments and discover a new achievement, rather than repeatedly accomplishing the same achievements.

We assume that the agent has no prior knowledge of the achievement graph, including the number of achievements and their dependencies. Additionally, the agent has no direct access to information about which achievements have been unlocked. Instead, the agent must infer this information indirectly from the reward signal it receives. Given this situation, our objective is to learn a generalizable policy $\pi : \mathcal{S} \to \mathcal{P}(\mathcal{A})$ that maximizes the expected return (*i.e.*, unlocks as many achievements as possible) across all environments of $\mathbb{M}$.

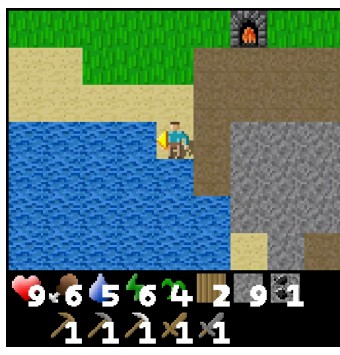

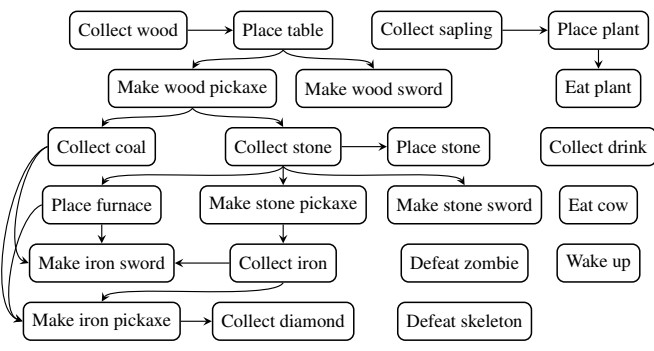

(a) Agent's observation          (b) Achievement graph with a hierarchical structure.

Figure 1: Overview of Crafter. Each environment is procedurally generated and partially observable. The objective is to unlock as many achievements as possible within a single episode.

## 2.2 Crafter environment

We primarily utilize the Crafter environment as a benchmark to assess the capabilities of an agent in solving MDPs with hierarchical achievements [19]. Crafter is an open-world survival game with 2D visual inputs, drawing inspiration from the popular 3D game Minecraft [17]. This is optimized for research purposes, with fast and straightforward environment interactions and clear evaluation metrics. The game consists of procedurally generated environments with varying world map layouts, terrain types, resource placements, and enemy spawn locations, each of which is uniquely determined by an integer seed. An agent can only observe its immediate surroundings as depicted in Figure 1a, which makes Crafter partially observable and thus challenging. To survive, the agent must acquire a variety of skills, including exploring the world map, gathering resources, building tools, placing objects, and defending against enemies. The game features a set of 22 hierarchical achievements that the agent can unlock by completing specific prerequisites, as illustrated in Figure 1b. For instance, to make a wood pickaxe, the agent needs to collect wood, place a table, and stand nearby. This achievement structure is designed to require the agent to learn and utilize a wide range of skills to accomplish increasingly challenging achievements, such as crafting iron tools and collecting diamonds.

## 2.3 Proximal policy optimization

PPO is one of the most successful model-free policy gradient algorithms due to its simplicity and effectiveness [39]. PPO learns a policy $\pi_\theta : \mathcal{S} \to \mathcal{P}(\mathcal{A})$ and a value function $V_\theta : \mathcal{S} \to \mathbb{R}$, which are parameterized by neural networks. During training, PPO first collects a new episodes $\mathcal{T}$ using the policy $\pi_{\theta_{old}}$ immediately prior to the update step. Subsequently, PPO updates the policy network using these episodes for several epochs to maximize the clipped surrogate policy objectives given by

$$J_\pi(\theta) = \mathbb{E}_{(s_t, a_t) \sim \mathcal{T}} \left[ \min \left( \frac{\pi_\theta(a_t \mid s_t)}{\pi_{\theta_{old}}(a_t \mid s_t)} \hat{A}_t, \text{clip} \left( \frac{\pi_\theta(a_t \mid s_t)}{\pi_{\theta_{old}}(a_t \mid s_t)}, 1 - \epsilon, 1 + \epsilon \right) \hat{A}_t \right) \right],$$

where $\hat{A}_t$ is the estimated advantage computed by generalized advantage estimate (GAE) [38]. PPO simultaneously updates the value network to minimize the value objective given by

$$J_V(\theta) = \mathbb{E}_{s_t \sim \mathcal{T}} \left[ \frac{1}{2} \left( V_\theta(s_t) - \hat{V}_t \right)^2 \right],$$

where $\hat{V}_t = \hat{A}_t + V_{\theta_{old}}(s_t)$ is the bootstrapped value function target.

In image-based RL, it is common practice to optimize the policy and value networks using a shared network architecture [13, 48]. An image observation is first passed through a convolutional encoder $\phi_\theta : \mathcal{S} \to \mathbb{R}^h$ to extract a state representation, which is then fed into linear heads to compute the policy and value function. Sharing state representations between the policy and value networks is crucial to improving the performance of agents in high-dimensional state spaces. However, relying solely on policy and value optimization to train the encoder can lead to suboptimal state representations, particularly in procedurally generated environments [9]. To address this issue, recent studies introduce an auxiliary training phase alongside the policy and value optimization that trains the encoder with auxiliary value or self-supervised objectives [9, 32].

# 3 Motivation

## 3.1 PPO is a strong baseline for hierarchical achievements

Despite being a simple and ubiquitous algorithm, PPO is less utilized than model-based or hierarchical approaches for solving MDPs with hierarchical achievements. This is due to the fact that PPO does not have an explicit component for long-term planning or reasoning, which is believed to be essential for solving hierarchical tasks. However, a recent study has shown that a PPO-based algorithm is also successful in solving hierarchical achievement on the Minecraft environment, albeit with the aid of pre-training on human video data [4].

Based on this observation, we first investigate the effectiveness of PPO in solving hierarchical achievements on Crafter without pre-training. We adopt the recent implementation practices proposed in Andrychowicz et al. [2], Baker et al. [4]. Concretely, we modify the default ResNet architecture in IMPALA as follows [14]:

- **Network size**: We increase the channel size from [16, 32, 32] to [64, 128, 128] and the hidden size from 256 to 1024.
- **Layer normalization**: We add layer normalization before each dense or convolutional layer [3].
- **Value normalization**: We keep a moving average for the mean and standard deviation of the value function targets and update the value network to predict the normalized targets.

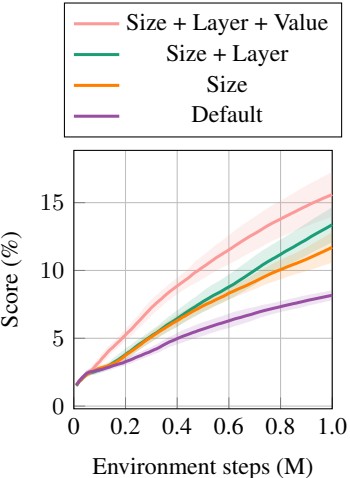

Figure 2: Score curves of PPO.

We train the modified PPO on Crafter for 1M environment steps and evaluate the success rates for unlocking achievements (please refer to Section 5.1 for the evaluation). Figure 2 shows that the slight modification in implementing PPO significantly improves the performance, increasing the score from 8.17 to 15.60. Notably, this outperforms the current state-of-the-art DreamerV3, which achieves a score of 14.77.

## 3.2 Representation analysis of PPO

Since Crafter environments are procedurally generated, simply memorizing successful episodes is insufficient for achieving high performance. We hypothesize that the PPO agent acquires knowledge beyond mere memorization of action sequences, possibly including information about achievements. To validate this, we analyze the learned latent representations of the encoder, as inspired by Wijmans et al. [46]. Specifically, we collect a batch of episodes using an expert policy and subsample a set of states for training. For each state $s$ in the training set, we freeze its latent representation $\phi_\theta(s)$ from the encoder. Subsequently, we train a linear classifier using this representation as input to predict the very next achievement unlocked in the episode containing $s$. Finally, we evaluate the classification accuracy and the prediction confidence of the ground-truth labels on a held-out test set. The detailed experimental settings are provided in Appendix A.

Surprisingly, PPO achieves a nontrivial accuracy of 44.9% in the 22-way classification. However, Figure 3 shows that the prediction outputs lack confidence with a median value of 0.240. This suggests that the learned representations of the PPO encoder are not strongly correlated with the next achievement to be unlocked and the agent may struggle to generate optimal action sequences towards a specific goal.

This finding warrants providing additional guidance to the encoder for predicting the next achievements with high confidence. However, since the agent has no access to the achievement labels, it is challenging to guide the agent in a supervised fashion. Therefore, it is necessary to explore alternative approaches to guide the agent toward predicting the next achievements.

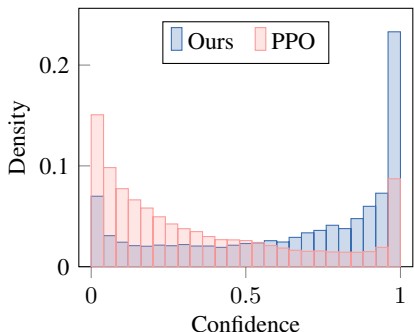

Figure 3: Histogram for the confidence of next achievement prediction.

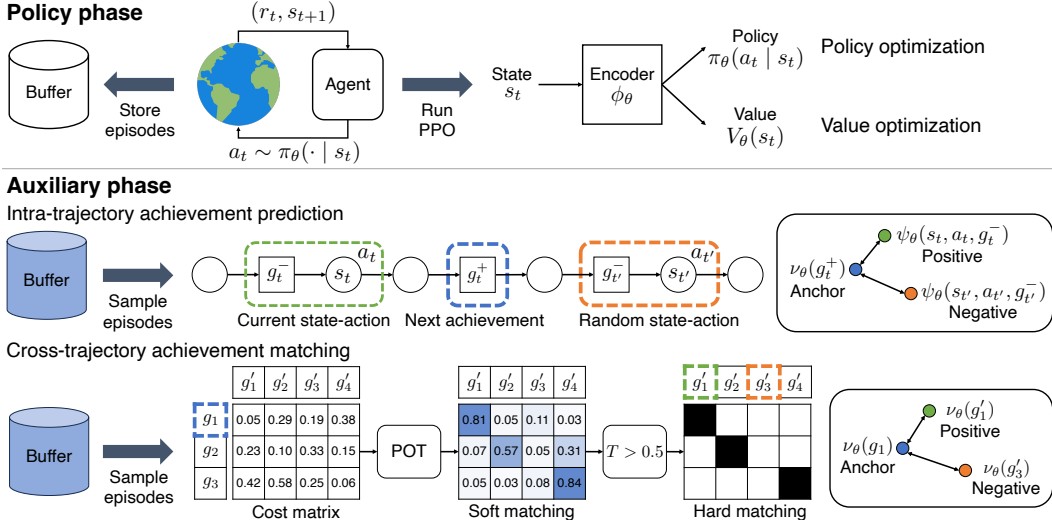

Figure 4: Illustration of achievement distillation.

# 4 Contrastive learning for achievement distillation

In this section, we introduce a new self-supervised learning method that works alongside RL training to guide the encoder in predicting the next achievement to be unlocked. This approach distills relevant information about discovered achievements from episodes collected during multiple policy updates into the encoder via contrastive learning. This method consists of two key components:

- **Intra-trajectory achievement prediction**: Within an episode, this maximizes the similarity in the latent space between a state-action pair and its corresponding next achievement.

- **Cross-trajectory achievement matching**: Between episodes, this maximizes the similarity in the latent space for matched achievements.

For ease of notation, we denote the sequence of unlocked achievements within an episode as $(g_i)_{i=1}^m$ and their corresponding timesteps as $(t_i)_{i=1}^m$, where each achievement $g_i$ is defined by a transition tuple $(s_{t_i}, a_{t_i}, s_{t_i+1})$. For each timestep $t$, we represent the very next achievement as $g_t^+ = g_u$, where $u = \min\{i \mid t \leq t_i\}$, and the very previous achievement as $g_t^- = g_l$, where $l = \max\{i \mid t > t_i\}$.

## 4.1 Intra-trajectory achievement prediction

Given a state-action pair $(s_t, a_t)$ and its corresponding next achievement $g_t^+$ within an episode $\tau$, we train the encoder $\phi_\theta$ to produce similar representations for them through contrastive learning [34, 22]. Specifically, we regard $g_t^+$ as the anchor and $(s_t, a_t)$ as the positive. We also randomly sample another state-action pair $(s_{t'}, a_{t'})$ from the same episode to serve as the negative. Subsequently, we obtain the normalized representations of the anchor, positive, and negative, denoted as $\nu_\theta(g_t^+)$, $\psi_\theta(s_t, a_t)$, and $\psi_\theta(s_{t'}, a_{t'})$, respectively. Finally, we minimize the following contrastive loss to maximize the cosine similarity between the anchor and positive representations while minimizing the cosine similarity between the anchor and negative representations:

$$L_{\text{pred}}(\theta) = -\mathbb{E}_{\substack{(s_t, a_t) \sim \tau \\ (s_{t'}, a_{t'}) \sim \tau}} \left[ \log \left( \frac{\exp(\psi_\theta(s_t, a_t)^\top \nu_\theta(g_t^+)/\lambda)}{\exp(\psi_\theta(s_t, a_t)^\top \nu_\theta(g_t^+)/\lambda) + \exp(\psi_\theta(s_{t'}, a_{t'})^\top \nu_\theta(g_t^+)/\lambda)} \right) \right],$$

where $\lambda > 0$ is the temperature parameter.

To obtain the state-action representation $\psi_\theta(s_t, a_t)$, we calculate the latent representation of the state $\phi_\theta(s_t)$ from the encoder and concatenate it with the action $a_t$ using a FiLM layer [35]. The resulting vector is then passed through an MLP layer and normalized. To obtain the achievement representation $\nu_\theta(g_t^+)$, we simply calculate the residual of the latent representations of the two consecutive states from the encoder and normalize it, as motivated by Nair et al. [33].

While this contrastive objective encourages the encoder to predict the next achievement in the latent space, it can potentially lead to distortions in the policy and value networks due to the changes in the encoder. To address this, we jointly minimize the following regularizers to preserve the outputs of the policy and value networks, following the practice in Moon et al. [32]:

$$R_\pi(\theta) = \mathbb{E}_{s_t \sim \tau} \left[ D_{\mathrm{KL}} \left( \pi_{\theta_{\mathrm{old}}}(\cdot \mid s_t) \parallel \pi_\theta(\cdot \mid s_t) \right) \right], \quad R_V(\theta) = \mathbb{E}_{s_t \sim \tau} \left[ \frac{1}{2} \left( V_\theta(s_t) - V_{\theta_{\mathrm{old}}}(s_t) \right)^2 \right],$$

where $\pi_{\theta_{\mathrm{old}}}$ and $V_{\theta_{\mathrm{old}}}$ are the policy and value networks immediately prior to the contrastive learning, respectively and $D_{\mathrm{KL}}$ denotes the KL divergence.

## 4.2 Cross-trajectory achievement matching

Since Crafter environments are procedurally generated, the achievement representations learned solely from intra-trajectory information may include environment-specific features that limit generalization. To obtain better achievement representations, we leverage the common achievement structure shared across all episodes.

We first match the sequences of unlocked achievements from two different episodes in an unsupervised fashion. Given two achievement sequences $\mathbf{g} = (g_i)_{i=1}^m$ and $\mathbf{g}' = (g_j')_{j=1}^n$, we define the cost matrix $M \in \mathbb{R}^{m \times n}$ as the cosine distance between the achievement representations:

$$M_{ij} = 1 - \nu_\theta(g_i)^\top \nu_\theta(g_j').$$

Subsequently, we regard these two sequences as discrete uniform distributions and compute a soft-matching $T \in \mathbb{R}^{m \times n}$ between them using partial optimal transport, which can be solved by

$$T = \arg\min_{T \geq 0} \langle T, M \rangle + \alpha \sum_{i=1}^m \sum_{j=1}^n T_{ij} \log T_{ij}$$

$$\text{subject to } T\mathbf{1} \leq \mathbf{1}, \ T^\top \mathbf{1} \leq \mathbf{1}, \ \mathbf{1}^\top T^\top \mathbf{1} = \min\{m, n\},$$

where $\alpha > 0$ is the entropic regularization parameter [5]. Here, we set the total amount of probability mass to be transported to the minimum length of the two sequences for simplicity. However, some unlocked achievements in one sequence may not exist in the other sequence, and therefore should not be transported. In this case, the optimal amount of probability mass to be transported should be less than the minimum length. To address this, we compute the conservative hard matching $T^*$ from $T$ by thresholding the probabilities less than 0.5 (*i.e.*, $T^* = \mathbb{1}[T > 0.5]$). Note that this also encourages each achievement to be matched to at most one other achievement. We provide examples of matching results in Appendix B.

We train the encoder to produce similar representations for the matched achievements according to $T^*$ through contrastive learning. Specifically, suppose that the $i$th achievement of the source sequence $g_i$ is matched with the $k$th achievement of the target sequence $g_k'$. Then, we consider $g_i$ as the anchor and $g_k'$ as the positive. We also randomly sample another achievement $g_j'$ from the target sequence to serve as the negative. Subsequently, we obtain the normalized representations of these achievements. Finally, we minimize the following contrastive loss to maximize the cosine similarity between the anchor and positive representations while minimizing the cosine similarity between the anchor and negative representations:

$$L_{\mathrm{match}}(\theta) = -\mathbb{E}_{g_i \sim \mathbf{g}, g_j' \sim \mathbf{g}'} \left[ \log \left( \frac{\exp(\nu_\theta(g_i)^\top \nu_\theta(g_k')/\lambda)}{\exp(\nu_\theta(g_i)^\top \nu_\theta(g_k')/\lambda) + \exp(\nu_\theta(g_i)^\top \nu_\theta(g_j')/\lambda)} \right) \right],$$

As in Section 4.1, we jointly minimize the policy and value regularizers to prevent distortions.

## 4.3 Achievement representation as memory

We further utilize the achievement representations learned in Sections 4.1 and 4.2 as memory for the policy and value networks. Specifically, given a state $s_t$ and its corresponding previous achievement $g_t^-$, we concatenate the latent state representation $\phi_\theta(s_t)$ with the previous achievement representation $\nu_\theta(g_t^-)$. The resulting vector is then fed into the policy and value heads to output an action distribution and value estimate, respectively.

We also utilize the previous achievement for the achievement prediction task in Section 4.1. Given a state-action pair $(s_t, a_t)$ and its corresponding previous achievement $g_t^-$, we concatenate the latent state representation $\phi_\theta(s_t)$ from the encoder with $a_t$ and $\nu_\theta(g_t^-)$. The resulting vector is then fed into an MLP layer and normalized to obtain the representation $\psi_\theta(s_t, a_t, g_t^-)$ for the next achievement prediction. This is interpreted as learning forward dynamics in the achievement space.

## 4.4 Integration with RL training

We integrate the contrastive learning method proposed in Sections 4.1 to 4.3 with PPO training by introducing two alternating phases, the policy and auxiliary phases. During the policy phase, which is repeated several times, we update the policy and value networks using newly-collected episodes and store them in a buffer. During the auxiliary phase, we update the encoder to optimize the contrastive objectives in conjunction with the policy and value regularizers using all episodes in the buffer. We call this auxiliary learning *achievement distillation*. The illustration and pseudocode are presented in Figure 4 and Algorithm 1, respectively.

---

**Algorithm 1** PPO with achievement distillation

---

**Require:** Policy network $\pi_\theta$, value network $V_\theta$
1: **for** phase $= 1, 2, \ldots$ **do**
2:      Reset the buffer $\mathcal{B}$
3:      **for** iter $= 1, 2, \ldots, N_\pi$ **do**                                   ▷ PPO training
4:          Collect episodes $\mathcal{T}$ using $\pi_\theta$ and add them to $\mathcal{B}$
5:          **for** epoch $= 1, 2, \ldots, E_\pi$ **do**
6:              Optimize $J_\pi(\theta)$ and $J_V(\theta)$ using $\mathcal{T}$
7:          **end for**
8:      **end for**
9:      $\pi_{\theta_{\text{old}}} \leftarrow \pi_\theta, V_{\theta_{\text{old}}} \leftarrow V_\theta$
10:     **for** iter $= 1, 2, \ldots, E_{\text{aux}}$ **do**                            ▷ Achievement distillation
11:         Optimize $L_{\text{pred}}(\theta)$, $R_\pi(\theta)$, and $R_V(\theta)$ using $\mathcal{B}$
12:         Optimize $L_{\text{match}}(\theta)$, $R_\pi(\theta)$, and $R_V(\theta)$ using $\mathcal{B}$
13:      **end for**
14: **end for**

---

# 5 Experiments

## 5.1 Experimental setup

To assess the effectiveness of our method in discovering hierarchical achievements, we train the agent on Crafter for 1M environment steps and evaluate its performance, following the protocol in Hafner [19]. We measure the success rates for all 22 achievements across all training episodes as a percentage and calculate their geometric mean to obtain our primary evaluation score[2]. Note that the geometric mean prioritizes unlocking challenging achievements. We also measure the episode reward, which indicates the number of achievements unlocked within a single episode, and report the average across all episodes within the most recent 100K environment steps. We conduct 10 independent runs using different random seeds for each experimental setting and report the mean and standard deviation. The code can be found at `https://github.com/snu-mllab/Achievement-Distillation`.

We compare our method with our backbone algorithm PPO and four baseline methods that have been previously evaluated on Crafter in other research work: DreamerV3, LSTM-SPCNN, MuZero + SPR, and SEA [21, 44, 45, 49]. DreamerV3 is a model-based algorithm that has achieved state-of-the-art performance on Crafter without any pre-training. LSTM-SPCNN is a model-free algorithm based on PPO that employs a recurrent and object-centric network to improve performance on Crafter. MuZero + SPR is a model-based algorithm that has demonstrated state-of-the-art performance on Crafter by utilizing unsupervised pre-training. SEA is a hierarchical algorithm based on IMPALA that employs

---

[2]The score is computed by $S = \exp\left(\frac{1}{N} \sum_{i=1}^{N} \ln(1 + s_i)\right) - 1$, where $s_i \in [0, 100]$ is the success rate of the $i$th achievement and $N = 22$.

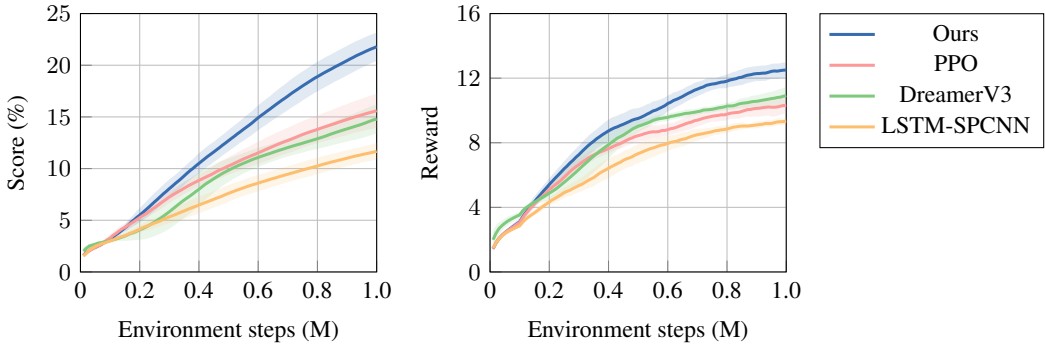

Figure 5: Score and reward curves.

a high-level planner to discover achievements on Crafter. We provide the implementation details and hyperparameters in Appendix C.

## 5.2 Crafter results

Table 1 and Figure 5 present the Crafter scores and rewards obtained by our method and the baselines. Our method outperforms all the baselines trained from scratch in both the metrics by a considerable margin, with a score of 21.79% and a reward of 12.60. Notably, our method exhibits superior score performance compared to MuZero + SPR, which utilizes pre-collected exploratory data and employs a computationally expensive tree search algorithm for planning, while achieving comparable rewards.

Figure 6 shows the individual success rates for all 22 achievements of our method and two successful baselines, DreamerV3 and LSTM-SPCNN. Remarkably, our method outperforms the baselines in unlocking challenging achievements. For instance, our method collects iron with a probability over 3%, which is 20 times higher than DreamerV3. This achievement is extremely challenging due to its scarcity on the map and the need for wood and stone tools. Moreover, our method crafts iron tools with a probability of approximately 0.01%, which is not achievable by either of the baselines. Finally, our method even succeeds in collecting diamonds, the most challenging task, on individual runs.

Table 1: Scores and rewards. MuZero + SPR[†] denotes the results replicated from the original paper.

|  | Method | Parameters | Score (%) | Reward |
|---|---|---|---|---|
|  | Human Expert | - | $50.5 \pm 6.8$ | $14.3 \pm 2.3$ |
|  | Ours | 9M | $\mathbf{21.79 \pm 1.37}$ | $\mathbf{12.60 \pm 0.31}$ |
| From scratch | PPO | 4M | $15.60 \pm 1.66$ | $10.32 \pm 0.53$ |
|  | DreamerV3 | 201M | $14.77 \pm 1.42$ | $10.92 \pm 0.53$ |
|  | LSTM-SPCNN | 135M | $11.67 \pm 0.80$ | $9.34 \pm 0.23$ |
|  | MuZero + SPR[†] | 54M | $4.4 \pm 0.4$ | $8.5 \pm 0.1$ |
|  | SEA | 1.5M | $1.22 \pm 0.13$ | $0.63 \pm 0.08$ |
| Pre-training | MuZero + SPR[†] | 54M | $16.4 \pm 1.5$ | $12.7 \pm 0.4$ |

## 5.3 Model size analysis

We compare the model sizes between our method and the baselines. As shown in Table 1, our method achieves better performance with fewer parameters. In particular, our method only requires 4% of the parameters used by DreamerV3. Note that while our method has twice as many parameters as PPO, most of this increase is due to the networks used for the auxiliary learning, which are not utilized during inference. Additionally, we test our method with a smaller model by reducing the channel size to [16, 32, 32] and the hidden dimension to 256, resulting in a total of 1M parameters. Notably, it still outperforms the baselines with a score of 17.07%.

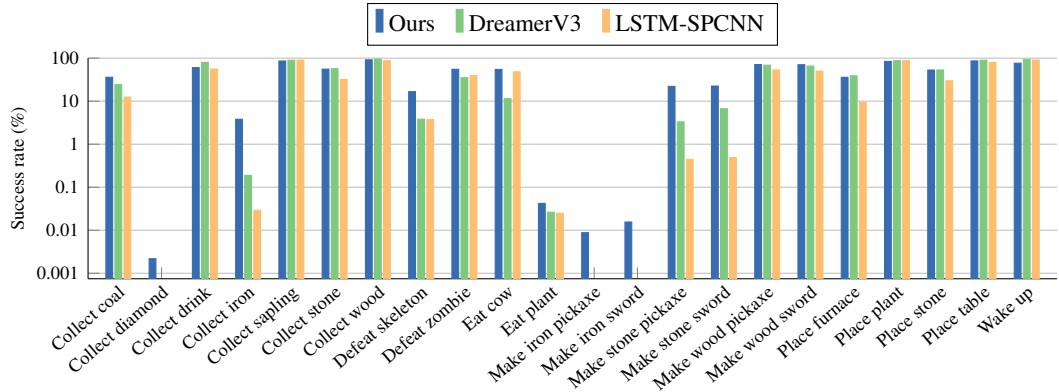

Figure 6: Individual success rates for all achievements.

## 5.4 Representation analysis of achievement distillation

To validate whether our method induces the encoder to have better representations for predicting the next achievements, we conduct an analysis of the latent representations of the encoder using the same approach as described in Section 3. Our method achieves a classification accuracy of 73.6%, which is a 28.7%p increase compared to PPO. Furthermore, Figure 3 demonstrates that our method produces predictions with significantly higher confidence than PPO, with a median value of 0.752.

## 5.5 Ablation studies

We conduct ablation studies to evaluate the individual contribution of our proposed method, intra-trajectory achievement prediction (I), cross-trajectory achievement matching (C), and memory (M). Table 2 shows that while intra-trajectory achievement prediction is the most significant contributor, cross-trajectory achievement matching and memory also play important roles in improving the performance of our method.

Table 2: Ablation studies.

| I | C | M | Score (%) |
|---|---|---|---|
| ✗ | ✗ | ✗ | $15.60 \pm 1.66$ |
| ✓ | ✗ | ✗ | $19.02 \pm 1.65$ |
| ✓ | ✓ | ✗ | $20.36 \pm 1.79$ |
| ✓ | ✓ | ✓ | $\mathbf{21.79 \pm 1.37}$ |

## 5.6 Extension to value-based algorithms

In our previous experiments, we use the on-policy policy gradient algorithm, PPO, as our backbone RL algorithm. To assess the adaptability of our method to other RL paradigms, we extend our experiments to the popular off-policy value-based algorithm, QR-DQN, introducing a slight modification [11]. Specifically, we employ Huber quantile regression to preserve the Q-network's output distribution in alignment with the value function optimization in QR-DQN. We train the agent on Crafter for 1M environment steps and evaluate its performance. Notably, our method also proves effective for value-based algorithms, elevating the score from 4.14 to 8.07. The detailed experimental settings and results are provided in Appendix D.

## 5.7 Application to other environments

To evaluate the broad applicability of our method to diverse environments, we conduct experiments on two additional benchmarks featuring hierarchical achievements: Procgen Heist and MiniGrid [8, 7]. Heist is a procedurally generated environment whose goal is to steal a gem hidden behind a sequence of blue, green, and red locks. To open each lock, an agent must collect a key with the corresponding color. Heist introduces another challenge, given that the color of wall and background can vary between environments, whereas Crafter maintains fixed color patterns for its terrains. Additionally, we create a customized a door-key environment using MiniGrid to evaluate the effectiveness of our method on a deeper achievement graph. Notably, our method significantly improves the performance of PPO in Heist, increasing the score from 29.6 to 71.0. Our method also outperforms PPO in the MiniGrid environment by a substantial margin, elevating the score from 3.33 to 8.04. The detailed experimental settings and results can be found in Appendix E.

# 6 Related work

**Discovering hierarchical achievements in RL**    One major approach to this problem is model-based algorithms. DreamerV3 learns a world model that predicts future states and rewards and trains an agent using imagined trajectories generated by the model [21]. While achieving superior performance on Crafter, it requires more than 200M parameters. MuZero + SPR trains a model-based agent with a self-supervised task of predicting future states [37, 45]. However, it relies on pre-training with 150M environment steps collected via RND to improve its performance on Crafter [6].

Another approach is hierarchical algorithms, while many of these methods are only tested on grid-world environments [43, 10]. HAL introduces a classifier that predicts the next achievement to be done and uses it as a high-level planner [10]. However, it relies on prior information on achievements, such as what achievement has been completed, and does not scale to the high-dimensional Crafter. SEA reconstructs the Crafter achievement graph using 200M offline data collected from a pre-trained IMPALA policy and employs a high-level planner on the graph [49]. However, it requires expert data to fully reconstruct the graph and has not been tested on a sample-efficient regime.

There are only a few studies that have explored model-free algorithms. LSTM-SPCNN uses an LSTM and a size-preserving CNN to improve the performance of PPO on Crafter [23, 26, 44]. However, it requires 135M parameters and the performance gain is modest compared to PPO with a CNN.

**Representation learning in RL**    There has been a large body of work on representation learning for improving sample efficiency on a single environment [25, 47, 40], or generalization on procedurally generated environments [29, 30, 32]. However, representation learning for discovering hierarchical achievements has little been explored. A recent study has evaluated the performance of the previously developed representation learning technique SPR on Crafter, but the results are not promising [40, 45].

While widely used in other domains, optimal transport has recently garnered attention in RL [12, 15, 27]. However, the majority of the studies focus on imitation learning, where optimal transport is employed to match the behavior of an agent with offline expert data. In this work, we utilize optimal transport to obtain generalizable representations for achievements in the online setting.

# 7 Conclusion

In this work, we introduce a novel self-supervised method for discovering hierarchical achievements, named *achievement distillation*. This method distills relevant information about achievements from episodes collected during policy updates into the encoder and can be seamlessly integrated with a popular model-free algorithm PPO. We show that our proposed method is capable of discovering the hierarchical structure of achievements without any explicit component for long-term planning, achieving state-of-the-art performance on the Crafter benchmark using fewer parameters and data.

While we utilize only minimal information about hierarchical achievements (*i.e.*, the agent receives a reward when a new achievement is unlocked), one limitation of our work is that we have not evaluated the transferability of our method to an unsupervised agent without any reward. A promising future direction would be developing a representation learning method that can distinguish achievements in a fully unsupervised manner and combining it with curiosity-driven exploration techniques [41, 16].

## Acknowledgements

This work was partly supported by Institute of Information & Communications Technology Planning & Evaluation (IITP) grant funded by the Korea government (MSIT) (No. 2020-0-00882, (SW STAR LAB) Development of deployable learning intelligence via self-sustainable and trustworthy machine learning, 80%, and No. 2022-0-00480, Development of Training and Inference Methods for Goal-Oriented Artificial Intelligence Agents, 20%). This research was supported by a grant from KRAFTON AI. This material is based upon work supported by the Air Force Office of Scientific Research under award number FA2386-23-1-4047. Hyun Oh Song is the corresponding author.

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
