# Supplementary Material for Discovering Hierarchical Achievements in Reinforcement Learning via Contrastive Learning

## A    Representation analysis

To analyze the latent representations learned by PPO and our method, we construct a dataset using an expert policy. Specifically, we initially train the expert policy using our method with 1M environment steps. Note that this policy is trained with a different seed from the policies used for evaluation. Next, we collect a batch of episodes using the expert policy, resulting in a dataset containing 215,578 states. From this dataset, we subsample 50,000 states for the training set and 10,000 states for the test set.

For each method, we acquire the latent representations from the encoder for the training set and freeze them. We then train a linear classifier using these representations to predict the very next achievements. Each achievement is labeled from 0 to 21, representing the different possible achievements in Crafter. We optimize the classifier for 500 epochs using the Adam optimizer with a learning rate of 1e-3 [9]. Finally, we measure the classification accuracy and the prediction probability (*i.e.*, confidence) for the ground-truth label on the test set.

## B    Examples of cross-trajectory achievement matching

To demonstrate the effectiveness of cross-trajectory achievement matching, we provide an example of matching results for our method and PPO. We first collect two different episodes using an expert policy, following the same procedure outlined in Appendix A. Subsequently, we acquire the representations of the achievement sequences for each method. Ultimately, we perform the matching process between the sequences of achievement representations for each method.

Figure 1 visualizes the cosine distance between the achievement representations from the two episodes. Remarkably, our method exhibits a lower cosine distance between the same achievements compared to PPO. This highlights the effectiveness of cross-trajectory achievement matching in facilitating the learning of generalizable representations for achievements across different episodes.

Figure 2 illustrates the soft matching results computed using partial optimal transport [2]. Notably, the matching result of PPO contains inaccurate or unconfident matchings, while our method does not suffer from this issue. Consequently, the hard matching result of PPO exhibits inaccurate matchings, as depicted in Figure 3b. For instance, "Defeat zombie" in the first episode is matched with "Eat cow' in the second episode and "Collect wood" is not matched at all. In contrast, Figure 3a shows that our method successfully matches identical achievements between episodes. Furthermore, our method avoids matching achievements in one episode that do not exist in the other episode.

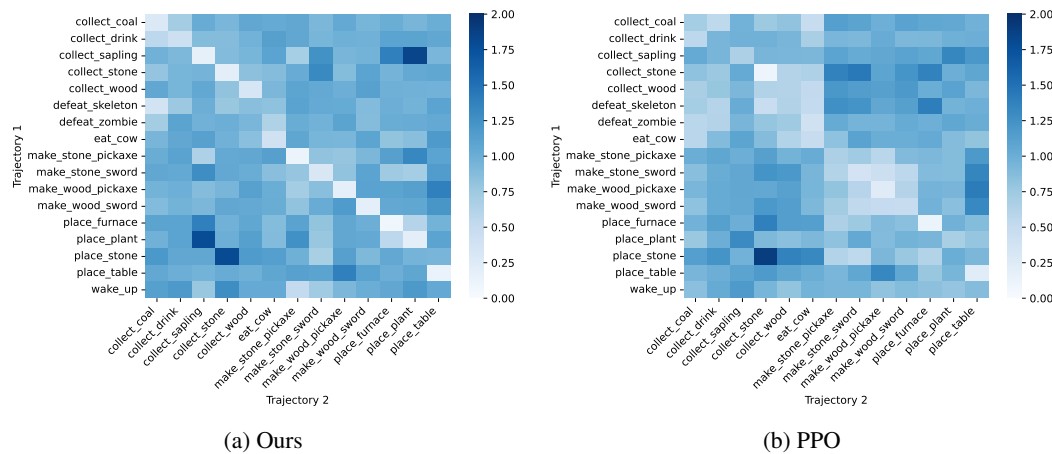

(a) Ours

(b) PPO

Figure 1: Cosine distance between achievement representations.

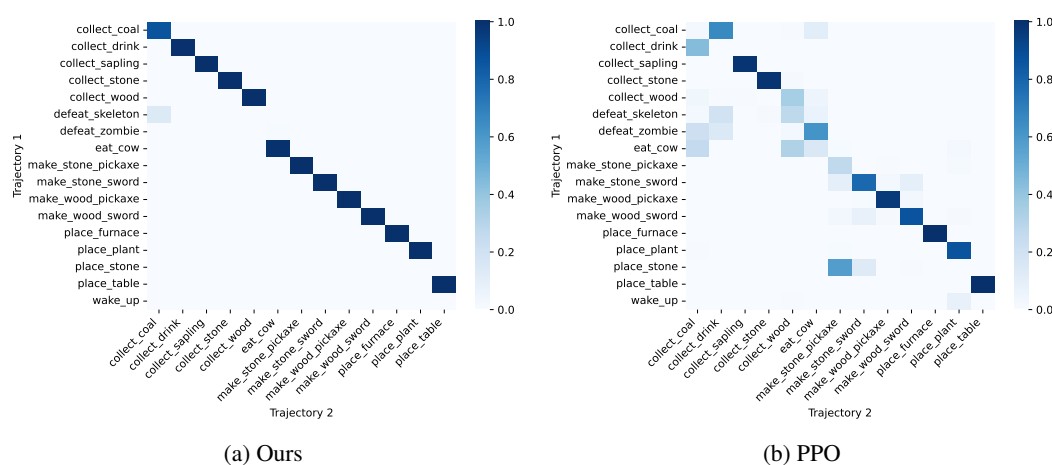

(a) Ours

(b) PPO

Figure 2: Soft matching via partial optimal transport.

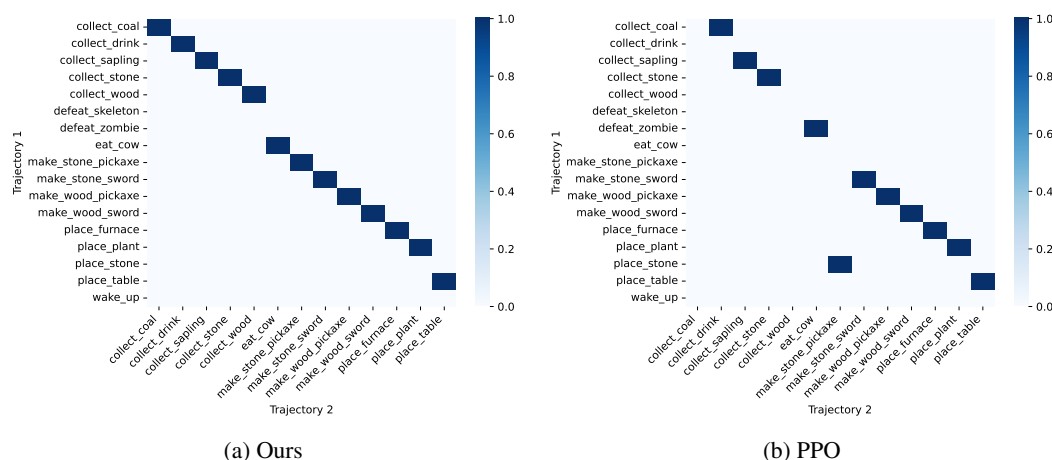

(a) Ours

(b) PPO

Figure 3: Hard matching via thresholding.

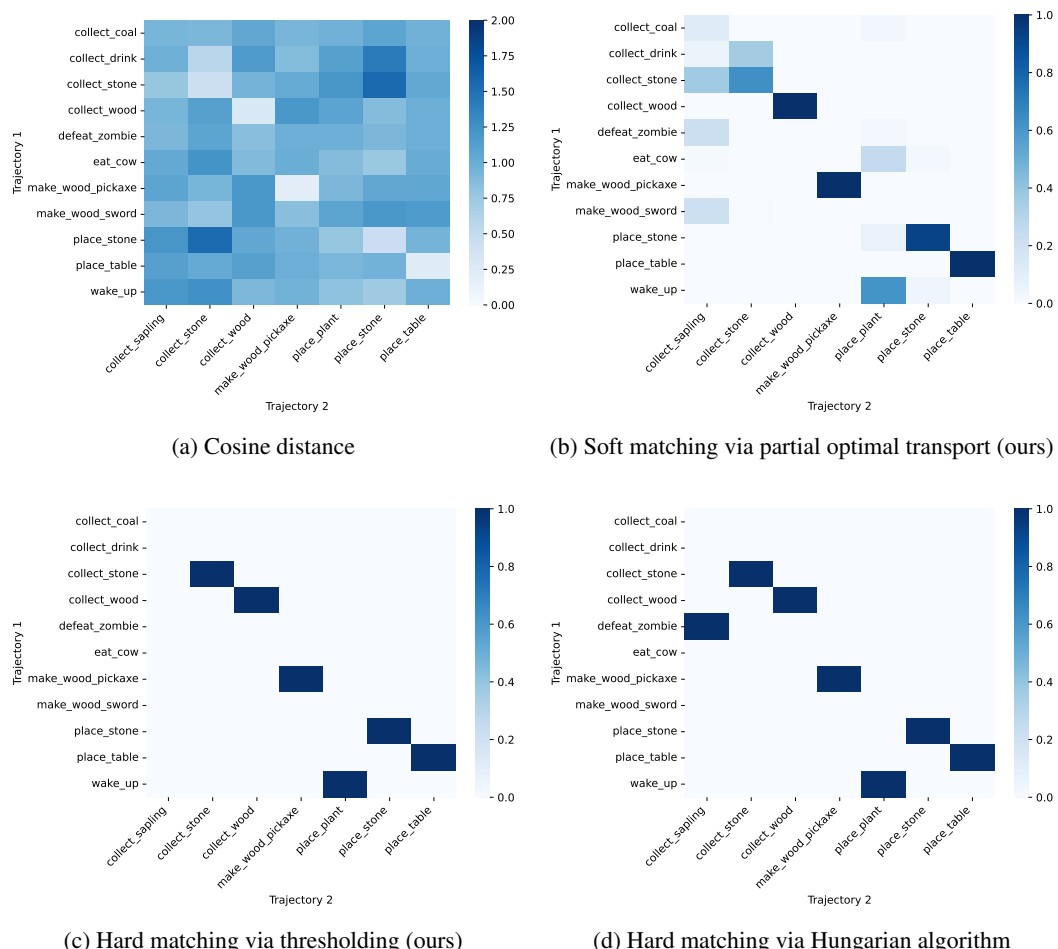

(a) Cosine distance

(b) Soft matching via partial optimal transport (ours)

(c) Hard matching via thresholding (ours)

(d) Hard matching via Hungarian algorithm

Figure 4: Matching results of our matching algorithm and the Hungarian algorithm.

We further explore the effectiveness of our matching algorithm that utilizes partial optimal transport followed by thresholding during the early stage of training. Specifically, we first collect two different episodes using a policy trained with our method for up to 100 epochs. Subsequently, we employ our matching algorithm to calculate the hard matching between the achievement sequences extracted from these episodes and compare it with the Hungarian algorithm, which is commonly used in bipartite graph matching [13].

Figure 4 provides an example of the matching results obtained using our matching algorithm and the Hungarian algorithm. In the case of the Hungarian algorithm, "Defeat zombie" in the first episode is incorrectly matched with "Collect sapling" in the second episode, as shown in Figure 4d. In contrast, our matching algorithm effectively avoids matching "Defeat zombie" in the first episode, as depicted in Figure 4c.

## C Experimental settings

### C.1 Computational resources

All experiments are conducted on an internal cluster, with each node consisting of two AMD EPYC 7402 CPUs, 500GB of RAM, and eight NVIDIA RTX 3090 GPUs. We utilize PyTorch as our primary deep learning framework [15].

### C.2 Implementation details and hyperparameters

**PPO**  Our implementation of PPO is based on the official code repository (`https://github.com/openai/Video-Pre-Training`) provided by Baker et al. [1]. Each input image has dimensions of $64 \times 64 \times 3$. The image is first processed with a ResNet encoder as proposed in IMPALA, which consists of three stacks with channel sizes of $[64, 64, 128]$ [4]. Each stack is composed of a $3 \times 3$ convolutional layer with a stride of 1, a $3 \times 3$ max pooling layer with a stride of 2, and two ResNet blocks as introduced in He et al. [6]. Subsequently, the output of the encoder is flattened into a vector of size 8192 and passed through two consecutive dense layers with output sizes of 256 and 1024, respectively. This resulting vector serves as the latent representation. Finally, the latent representation is fed into two independent dense layers, the policy and value heads. The policy head has an output size of 17 and generates a categorical distribution over the action space. The value head has an output size of 1 and produces a scalar value representing the value function. All weights are initialized using fan-in initialization, except for the policy and value heads [10]. The weights of the policy and value heads are initialized using orthogonal initialization with a gain of $0.01$ and $0.1$, respectively [17]. All biases are initialized to zero. We use ReLU as the activation function [14]. All network parameters are optimized using the Adam optimizer [9].

For PPO training, we slightly modify the hyperparameter setting from Stanić et al. [18]. Specifically, we decrease the number of mini-batches per epoch from 32 to 8 and the number of epochs per rollout from 4 to 3. These modified settings are commonly used in Cobbe et al. [3], Moon et al. [12]. Instead of employing the reward normalization technique proposed in the original PPO paper, we normalize the value function target with the mean and standard deviation estimated through an exponentially weighted moving average (EWMA), following the practice in Baker et al. [1]. We set the decay rate of the EWMA to 0.99. We provide the default values for the hyperparameters in Table 1.

Table 1: PPO hyperparameters.

| Hyperparameter | Value |
|---|---|
| Discount factor | 0.95 |
| GAE smoothing parameter | 0.65 |
| # timesteps per rollout | 4096 |
| # epochs per rollout | 3 |
| # mini-batches per epoch | 8 |
| Entropy bonus | 0.01 |
| PPO clip range | 0.2 |
| Reward normalization | No |
| EWMA decay rate | 0.99 |
| Learning rate | 3e-4 |
| Max grad norm | 0.5 |
| Value function coefficient | 0.5 |

**DreamerV3**  To reproduce the results, we utilize the official code repository (`https://github.com/danijar/dreamerv3`) provided by Hafner et al. [5]. We use the recommended hyperparameter setting from the original paper. Specifically, we set the model size to XL and the training ratio, which is the number of imagined steps per environment step, to 512. For more detailed information about the hyperparameter, please refer to Table 2.

**LSTM-SPCNN**  To reproduce the results, we utilize the official code repository (`https://github.com/astanic/crafter-ood`) provided by Stanić et al. [18]. Regarding the network architecture, a

Table 2: DreamerV3 hyperparameters.

| Hyperparameter | Value |
| --- | --- |
| GRU recurrent units | 4096 |
| CNN multiplier | 96 |
| Dense hidden units | 1024 |
| MLP layers | 5 |
| Training ratio | 512 |

size-preserving CNN is used as the image encoder [11]. This encoder consists of four convolutional layers, each with a kernel size of $5 \times 5$, a channel size of 64, and a stride of 1, and does not have any pooling layers. The encoder output is flattened into a vector of size 262144 and fed into a dense layer with an output size of 512, yielding the latent representation. For the policy network, the latent representation is first passed through an LSTM layer with a hidden size of 256 and then fed into a dense layer with an output size of 17, producing a categorical distribution over the actions [8]. For the value network, the latent representation is passed through two consecutive dense layers with output sizes of 256 and 1, respectively, to generate a scalar value representing the value function. For PPO training, we adopt the best hyperparameter setting from the original paper. Please refer to Table 3 for the default values for the hyperparameters.

Table 3: LSTM-SPCNN hyperparameters.

| Hyperparameter | Value |
| --- | --- |
| Discount factor | 0.95 |
| GAE smoothing parameter | 0.65 |
| # timesteps per rollout | 4096 |
| # epochs per rollout | 4 |
| # minibatches per epoch | 32 |
| Entropy bonus | 0.0 |
| PPO clip range | 0.2 |
| Reward normalization | No |
| Learning rate | 3e-4 |
| Max grad norm | 0.5 |
| Value function coefficient | 0.5 |

**MuZero + SPR**  We report the results replicated from the original paper since the official code has not been released yet [19]. It is worth noting that the paper uses an increased resolution for the input images, from $64 \times 64$ to $96 \times 96$, which can potentially result in improved performance compared to the original settings.

**SEA**  To evaluate its performance, we utilize the official code repository (`https://github.com/pairlab/iclr-23-sea`) provided by Zhou and Garg [20]. It is important to mention that the paper uses a modified version of Crafter, where the health mechanism and associated rewards are removed. This modification allows the agent to be immortal and explore the world map without any constraints, making the environment much easier than the original. Furthermore, the paper trains an agent using a substantial number of 500M environment steps. The paper uses 200M environment steps to train the IMPALA policy for offline data collection, followed by additional 300M environment steps to train the sub-policies for exploration.

To ensure a fair comparison with other methods, we reproduce the results by training an agent on the original Crafter benchmark using 1M environment steps. Specifically, we use 400K environment steps for training the IMPALA policy and 600K environment steps for training the sub-policies, maintaining the same ratio as the original setting. Regarding the network architecture, we keep it unchanged from the original implementation. However, it is worth noting that the original implementation employs a higher resolution of $84 \times 84$ for the input images, which can potentially improve the performance. We adopt the best hyperparameter setting for training from the original paper. Please refer to Table 4 for the default values for the hyperparameters.

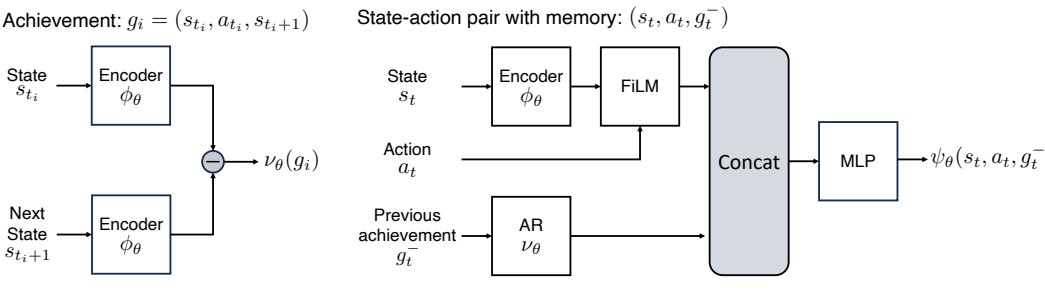

(a) Achievement representation    (b) State-action pair representation

Figure 5: Network architectures for the representations of (a) achievements and (b) state-action pairs.

Table 4: SEA hyperparameters.

| Hyperparameter | Value |
|---|---|
| Network architecture | CNN + LSTM |
| Hidden size | 256 |
| Learning rate | 2e-4 |
| Batch size | 32 |
| Unroll length | 80 |
| Gradient clipping | 40 |
| RMSProp $\alpha$ | 0.99 |
| RMSProp momentum | 0 |
| RMSProp $\epsilon$ | 0.01 |
| Discount factor | 0.99 |
| Reward normalization | Yes |
| # timesteps for IMPALA policy training | 400K |
| # timesteps for achievement classifier training | 100K |
| # timesteps for sub-policy training | 600K |

**Achievement distillation**   Our method is built upon the PPO implementation, utilizing the same network architecture and hyperparameters for PPO training. For achievement distillation, we compute the representation of an achievement $g_i = (s_{t_i}, a_{t_i}, s_{t_i+1})$ by computing the difference between the latent state representations from the encoder:

$$\nu_\theta(g_i) = \phi_\theta(s_{t_i+1}) - \phi_\theta(s_{t_i}),$$

which is then normalized. This process is illustrated in Figure 5a. We also compute the representation of a state-action pair with the previous achievement as memory $(s_t, a_t, g_t^-)$ as

$$\psi_\theta(s_t, a_t, g_t^-) = \text{MLP}_\theta(\text{Concat}(\text{FiLM}_\theta(\phi_\theta(s_t), a_t), \nu_\theta(g_t^-))),$$

as described in Figure 5b. Specifically, the latent state representation from the encoder is combined with the action using a FiLM Layer:

$$\text{FiLM}_\theta(\phi_\theta(s_t), a_t) = (1 + \eta_\theta(a_t))\phi_\theta(s_t) + \delta_\theta(a_t),$$

where $\eta_\theta$ and $\delta_\theta$ are two-layer MLPs, each with a hidden size of 1024 [16]. The resulting vector is then concatenated with the achievement representation and passed through a two-layer MLP with a hidden size of 1024, followed by normalization.

When optimizing the contrastive objectives for achievement prediction and achievement matching, we jointly optimize the policy and value regularizer with the policy regularizer coefficient $\beta_\pi = 1.0$ and the value regularizer coefficient $\beta_V = 1.0$. We compute a soft-matching between two achievement sequences using partial optimal transport with the entropic regularizer coefficient $\alpha = 0.05$. Note that these values are set without any hyperparameter search.

For the policy and auxiliary phases, we search for the number of policy phases per auxiliary phase within the range of $\{4, 8, 16\}$ and find $N_\pi = 8$ is optimal. Similarly, we sweep over different values for the number of epochs per auxiliary phase, considering values of $\{1, 3, 6\}$, and determine $E_{\text{aux}} = 6$ as the optimal choice. We provide the default values for the hyperparameter in Table 5.

Table 5: Achievement distillation hyperparameters.

| Hyperparameter | Value |
|---|---|
| Policy regularizer coefficient ($\beta_\pi$) | 1.0 |
| Value regularizer coefficient ($\beta_V$) | 1.0 |
| Entropic regularizer coefficient ($\alpha$) | 0.05 |
| # policy phases per auxiliary phase ($N_\pi$) | 8 |
| # epochs per auxiliary phase ($E_{\mathrm{aux}}$) | 6 |

## C.3 Environment details

**Observation space**    An agent receives an image observation of dimensions $64 \times 64 \times 3$. This image contains a local, agent-centric view of the world map and the inventory state of the agent, such as the quantities of resources and tools and the levels of health, food, water, and energy.

**Action space**    Crafter features a discrete 17-dimensional action space. The complete list of possible actions is provided in Table 6. The "Do" action encompasses activities including resource collection, consumption of food and water, and combat against enemies.

Table 6: Crafter action space.

| Index | Name |
|---|---|
| 0 | Noop |
| 1 | Move left |
| 2 | Move right |
| 3 | Move up |
| 4 | Move down |
| 5 | Do |
| 6 | Sleep |
| 7 | Place stone |
| 8 | Place table |
| 9 | Place furnace |
| 10 | Place plant |
| 11 | Make wood pickaxe |
| 12 | Make stone pickaxe |
| 13 | Make iron pickaxe |
| 14 | Make wood sword |
| 15 | Make stone sword |
| 16 | Make iron sword |

**Achievements**    Crafter consists of 22 achievements that the agent can unlock by satisfying specific requirements. The complete list of the achievements and their corresponding requirements is presented in Table 7.

Table 7: Crafter achievements and their requirements.

| Name | Requirements |
| --- | --- |
| Collect coal | Nearby coal; wood pickaxe |
| Collect diamond | Nearby diamond; iron pickaxe |
| Collect drink | Nearby water |
| Collect iron | Nearby iron; stone pickaxe |
| Collect sapling | None |
| Collect stone | Nearby stone; wood pickaxe |
| Collect wood | Nearby wood |
| Defeat skeleton | None |
| Defeat zombie | None |
| Eat cow | None |
| Eat plant | Nearby plant |
| Make iron pickaxe | Nearby table and furnace; wood, coal, and iron |
| Make iron sword | Nearby table and furnace; wood, coal, and iron |
| Make stone pickaxe | Nearby table; wood and stone |
| Make stone sword | Nearby table; wood and stone |
| Make wood pickaxe | Nearby table; wood |
| Make wood sword | Nearby table; wood |
| Place furnace | Stone |
| Place plant | Sapling |
| Place stone | Stone |
| Place table | Wood |
| Wake up | None |

# D  Extension to value-based algorithms

Our implementation of QR-DQN is derived from an open-source implementation (`https://github.com/Kaixhin/Rainbow`) of Rainbow [7]. We use the original hyperparameter settings for training. For the network architecture, we employ the ResNet encoder, consistent with our PPO implementation. We apply our contrastive learning method to the Q-network encoder prior to each target Q-network update. The default values for the hyperparameters are shown in Table 8. Our method significantly improves the performance of QR-DQN, as shown in Figure 6.

Table 8: QR-DQN hyperparameters.

| Hyperparameter | Value |
| --- | --- |
| Discount factor | 0.95 |
| Batch size | 64 |
| Replay frequency | 4 |
| Target update | 8000 |
| Learning rate | 6.25e-5 |
| Adam $\epsilon$ | 1.5e-4 |
| Max grad norm | 10 |
| Exploration strategy | $\epsilon$-greedy |

# E  Application to other environments

## E.1  Procgen Heist

Heist is a procedurally generated door-key environment, whose goal is to steal a gem after unlocking a sequence of blue, green, and red locks, as illustrated in Figure 7. To open each lock, an agent must collect a key with the corresponding color. We consider unlocking each lock and stealing a gem as achievements. To ensure closer alignment with Crafter, we slightly adjust the reward structure so that an agent receives a reward of 2 for opening each lock and a reward of 10 for successfully stealing a

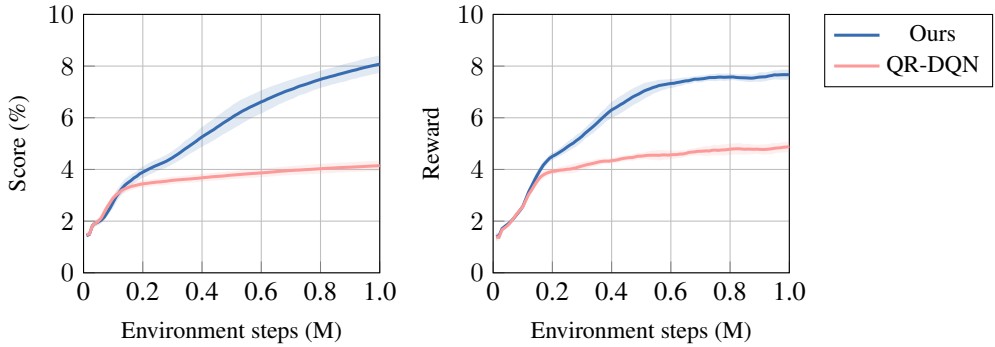

Figure 6: Crafter scores and rewards with QR-DQN algorithm.

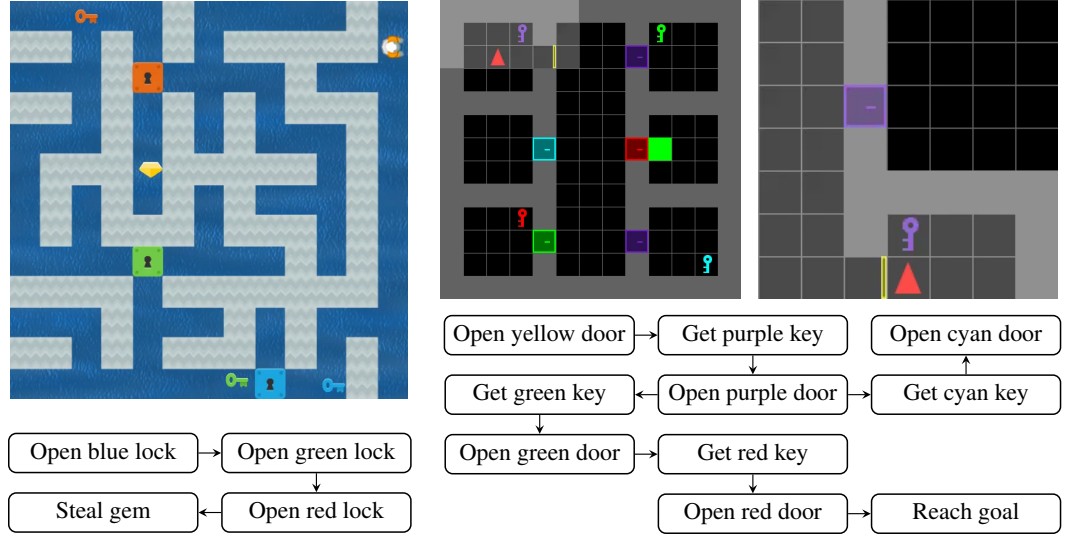

Figure 7: Overview of Procgen Heist. Each map features different layout and background color.

Figure 8: Overview of Custom MiniGrid environment. Each map consists of six rooms (left) and an agent observes its 7×7 surroundings (right).

gem. We train the agent in the "hard" difficulty mode for 25M environment steps and evaluate its performance in terms of the success rate of gem pilfering and the episode reward. Figure 9 shows that our method outperforms PPO by a significant margin throughout training.

## E.2  MiniGrid

The design of the custom MiniGrid environment takes inspiration by TreeMaze proposed in SEA [20]. An agent must sequentially unlock doors, find keys, and finally reach the green square, as depicted in Figure 8. The environment comprises a total of 10 achievements. An agent receives a reward of 1 for unlocking a new achievement, mirroring the reward structure in Crafter. We train an agent for 1M environment steps and evaluate its performance in terms of the geometric mean of success rates and the episode reward, following the same protocol as Crafter. Figure 10 demonstrates that our method outperforms PPO and showcases reduced variance across various training seeds.

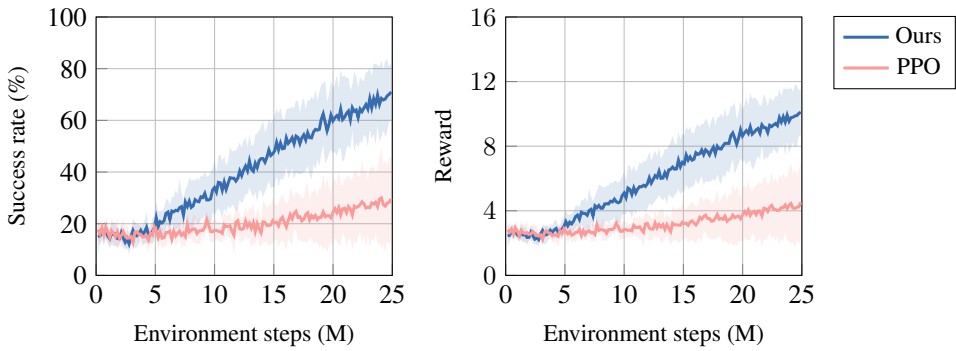

Figure 9: Procgen Heist success rate and reward curves.

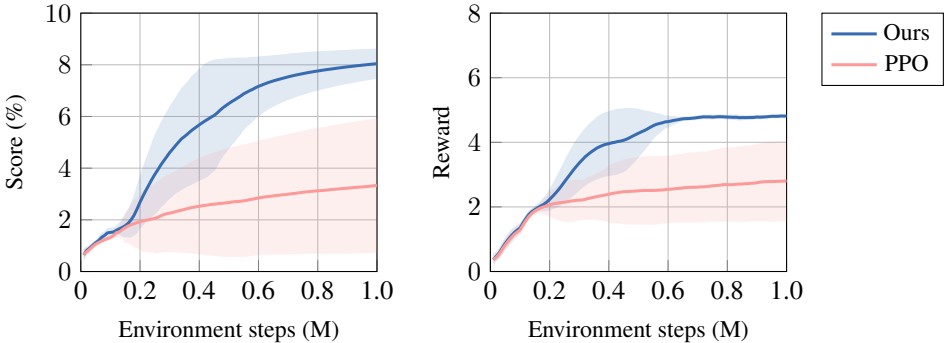

Figure 10: Minigrid score and reward curves.