# OpenReview forum: "Discovering Hierarchical Achievements in Reinforcement Learning via Contrastive Learning"
_NeurIPS.cc/2023/Conference — NeurIPS 2023 poster_

### Official Review · Reviewer_cpyF · 2023-06-26

**Soundness:** 2 fair
**Presentation:** 3 good
**Contribution:** 3 good
**Rating:** 6
**Confidence:** 3

**Summary:**

This paper focuses on the problem of sequential decision-making within a hierarchical framework, where tasks exhibit a hierarchical decomposition structure, and the agent does not possess any prior knowledge of the task dependency graph. In contrast to prior hierarchical approaches that directly model dependencies and utilize two-level policies for task resolution, it investigates an even more demanding scenario. Specifically, the agent lacks information about the unlocked achievements within each episode.

To tackle this challenge, the paper employs PPO as its backbone RL algorithm. Interestingly, it is found that the representations learned by PPO possess some ability to predict the next locked achievement, albeit with limited confidence. To further enhance this predictive capability, the paper proposes two contrastive loss mechanisms as representation learning objectives. These two mechanisms help the representation to be able to predict the agent’s next unlocked achievement (intra-trajectory contrastive loss) and also learn a representation of the achievement so that it is not capturing any environment-specific or spurious features (cross-trajectory matching).


**Strengths:**

1. The author proposes a novel self-supervised loss within the context of hierarchical decision-making, which could be easily integrated into the existing RL algorithm (PPO).
2. Good empirical performance boost compared with the previous strongest model-based RL algorithms (Dreamer-v3) on the Crafter environment.


**Weaknesses:**

My concerns is mostly around how relaxing the assumption of the agent's knowledge of its unlocked achievement during an episode. In particular, whether it is realistic in real applications, and how much does this affect the overall performance (in other words comparison to a  baseline method that does assume such prior knowledge). Please see my detailed comments in the Questions section.

**Questions:**

Here are my questions about the paper. Given the limited amount of time for rebuttal, I understand that it may not be possible to address all of the points with comprehensive additional experiments. But I would be happy to raise my score if my following concerns are adequately addressed.

1. **Assumptions of the problem setting**: The paper introduces an additional level of difficulty by assuming that the agent does not possess knowledge of the specific achievements unlocked throughout each episode. This assumption diverges from previous model-based and hierarchical approaches, which rely on such information. It would be helpful if the authors could provide a real-world example or scenario where relaxing this assumption becomes necessary. This would aid in understanding the practical applicability of such an assumption. From my perspective, this assumption appears weak, as it is difficult to envision an agent successfully completing an episode without awareness of the achievements attained during the process.

2. **Backbone RL algorithm**: The paper extensively discusses the effectiveness of the Proximal Policy Optimization (PPO) algorithm as the backbone for addressing tasks with hierarchical structures. The authors demonstrate how the representations learned by the PPO agent exhibit predictive abilities for the next achievement, as outlined in sections 3.1 and 3.2. However, it would be valuable to clarify whether these findings are specific to policy gradient methods or if they can be extended to value-based approaches as well. Understanding if the proposed contrastive loss mechanisms can be applied to value-based reinforcement learning algorithms would contribute to a more comprehensive evaluation of their potential.

3. **Baseline Comparison**: It would be insightful to compare the performance of the two proposed contrastive losses with an "oracle" objective, where the agent does possess prior knowledge of its unlocked achievements. For instance, for intra-trajectory achievement prediction, a simple 22-way classification could be employed as the loss, predicting the next achievement based on the current state-action pair. Similarly, for the cross-trajectory objective, minimizing the distance between representations of the same achievements could be considered. Such a comparison would facilitate a better understanding of the impact of relaxing the assumption regarding the agent's knowledge of achievements on the overall algorithm performance.

---

> ### Author Rebuttal · Authors · 2023-08-09
>
> We thank the reviewer for the valuable feedback. We appreciate the encouraging comments (“the author proposes a novel self-supervised loss within the context of hierarchical decision-making”, “good empirical performance boost compared with the previous strongest model-based RL algorithms”). We would like to address the questions and concerns of the reviewer, as presented below.
>
> ---
>
> **Q1: Assumption of the problem setting appears weak.**
>
> We wish to emphasize that our work focuses on developing an agent capable of discovering reusable skills and composing them to solve complex tasks in open-ended environments, as stated in the first paragraph of Section 1 in our main paper. We agree with your view that, for completing specific tasks with pre-defined subtask dependencies (e.g., making pasta with a detailed recipe), leveraging these dependency structures (e.g., providing the entire recipe) and incorporating subtask completion signals (e.g., acknowledging the completion of making the sauce) would be beneficial for training an agent [1, 2]. However, in open-ended environments where the final goal is not clearly defined and an agent faces the challenge of solving a myriad of tasks, pre-defining subtask dependencies for each task could be impractical. In such situations, an agent must explore environments, build a repertoire of skills, and combine these skills to solve tasks in an autonomous way.
>
> One realistic example is the game of Minecraft, where players are not provided with explicit instructions for survival, such as building shelters or crafting weapons. Instead, they must find their own survival strategies through exploration. Due to the unlimited number of potential survival strategies, defining specific subtask dependencies or providing subtask completion signals becomes infeasible. In this context, it is reasonable to assume that an agent receives a reward if they discover a new survival strategy. This approach aligns neatly with our reward assumption where an agent is granted a reward of 1 when unlocking a new achievement. It is important to note that the objective of Crafter is not solely about collecting diamonds. There is no pre-defined end goal and an agent is rewarded as it continually discovers and develops new survival skills.
>
> Thank you for raising this particular concern regarding the assumption. We appreciate your feedback and recognize the need to elucidate realistic scenarios where our assumption holds. We will revise the introduction section to provide a more in-depth explanation of the practical applicability of our assumption. Additionally, we will revisit lines 39 to 41 of the main paper and rewrite them to better articulate the potential limitations of the hierarchical approach in an open-ended world.
>
> **Q2: Application of contrastive learning to value-based methods.**
>
> Thank you for bringing this to our attention. We evaluate our contrastive learning method alongside a popular off-policy value-based algorithm QR-DQN and observe its strong performance. For more details on the experimental settings and results, please refer to General Response Q2.
>
> **Q3: It would be insightful to compare the performance of the two proposed contrastive losses with an "oracle" objective.**
>
> Thank you for your valuable suggestion. We train PPO agents with the oracle objectives on Crafter for 1M environment steps and compare their performance with our proposed contrastive objectives. Following your recommendation, we substitute the intra-trajectory contrastive prediction with a 23-way classification (22 achievements and 1 no achievement) using oracle labels. Additionally, we replace the cross-trajectory Wasserstein matching with the exact matching using oracle labels. To more precisely assess the impact of each objective, we choose not to employ the memory component, which is not necessarily required for implementing the intra- and cross-trajectory objectives.
>
> Tables 1 and 2 of the attached file present the performance of the oracle intra- and cross-trajectory objectives, respectively. Notably, the oracle cross-trajectory matching outperforms our approach. When it comes to the intra-trajectory prediction, however, there is no significant difference between the oracle and our method. It is important to highlight that the achievement label itself does not contain detailed information about the achievement. For example, the label indicating crafting a stone pickaxe omits essential information regarding the agent's requirement for wood and stone, as well as proximity to a crafting table. We hypothesize that predicting the next achievement in the latent space of the encoder, whose representations may encompass richer information about object location and inventory states, provides additional benefit over predicting achievement labels.
>
> ---
>
> Thank you again for your constructive comments, which help us to improve our paper’s quality. We hope that our answers address all the reviewer's points.
>
> **References**
>
> [1] Sungryull Sohn et al., Hierarchical Reinforcement Learning for Zero-shot Generalization with Subtask Dependencies, NeurIPS 2018. \
> [2] Robby Costales et al., Possibility Before Utility: Learning And Using Hierarchical Affordances, ICLR 2022.

---

> > ### Comment · Reviewer_cpyF · 2023-08-11
> > **Thank you!**
> >
> > I appreciate the additional experiments conducted by the authors, and they have adequately addressed my concerns. I would love to see the work accepted and shared with the community at large.

---

### Official Review · Reviewer_nAYF · 2023-07-02

**Soundness:** 3 good
**Presentation:** 3 good
**Contribution:** 3 good
**Rating:** 7
**Confidence:** 3

**Summary:**

This paper introduces achievement distillation, a representation learning method that is combined with PPO to obtain state-of-the-art results on the 2D crafter benchmark. First, the authors demonstrate that with simple hyper-parameter tweaks, the performance of vanilla PPO can be greatly improved. Next, detail the three components of achievement distillation which leverage achievement labels $g$. Intra-trajectory achievement prediction uses a contrastive objective to maximize the similarity between state action pairs and the next achievement. Cross-trajectory achievement matching maximizes the similarity between state-action pairs of the same achievement across episodes using optimal transport. Finally, they use the achievement representations as memory by concatenting the last achievement representation to the policy and value inputs. This results in a method that achieves high performance on the Crafter benchmark, especially for difficult to reach achievements.


**Strengths:**

The paper was generally easy to follow.

The experiments improving the performance of vanilla PPO were exciting! It’s great to see better tuned baselines. The improvements in PPO also naturally flowed into the introduction of achievement distillation.

Achievement distillation is an interesting form of representation learning, and to my knowledge, is novel.

The authors conducted experiments only on the Crafter environment. The results in this benchmark are compelling as it is extremely challenging, and as the authors highlight, achievement distillation can perform better on very hard to achieve tasks.

The authors additionally include ablation studies by subtracting components of their method.

I found the ideas in the paper to be easy to follow and straightforward in a good way.


**Weaknesses:**

The approach appears slightly overfit to the chosen Crafter benchmark, and the authors do not test achievement distillation in any other settings. Though the results on Crafter are compelling, it would be great if the authors could provide more examples and discussion of when this could be applied to other environments, and if so, how.

The comparison to baselines is a bit mis-leading, as other approaches, like DreamerV3, do not make use of the achievement labels necessary for achievement distillation. That being said, even PPO outperforms these baselines! I also found the log-scale on the axes of Figure 6 to be a bit confusing – is a success rate < 0.01% really significant?

I generally recommend acceptance of this work, primary drawbacks being its limited applicability to broader settings and evaluation on only one environment.



**Questions:**

Major:
Since crafter is partially observed, do you use a recurrent network?

Minor:
Line 141: outperforms dreamer: for the same number of environment steps? I would expected model-based to have higher sample efficiency, though it may have lower asymptotic performance.


**Limitations:**

The work is heavily designed for the crafter benchmark. While this is impressive, it would be good if the authors could include a discussion of where achievement distillation would be useful beyond just this setting. Where is it inapplicable? What are the challenges in implementing this?

---

> ### Author Rebuttal · Authors · 2023-08-09
>
> We thank the reviewer for the helpful feedback. We are encouraged by the reviewer’s positive comments (“the experiments improving the performance of vanilla PPO were exciting”, “the ideas in the paper are easy to follow and straightforward in a good way”). We would like to address the questions raised by the reviewer, as presented below.
>
> ---
>
> **Q1: More benchmarks. It would be great if the authors could provide more examples and discussion of when this could be applied to other environments.**
>
> Thank you for your suggestion. We conduct experiments on two additional benchmarks that feature hierarchical achievements: Procgen Heist and a custom MiniGrid environment. Please refer to the Global Response Q1 for more details on the benchmarks and our experimental results.
>
> **Q2: DreamerV3 does not make use of the achievement labels.**
>
> We first clarify that our method operates under the assumption that an agent has information only regarding when a new achievement is unlocked and does not utilize achievement labels, as detailed in Section 2.1 of our main paper. Additionally, this information can be easily retrieved from the reward signal, specifically at timesteps when rewards are 1. The sole distinction from DreamerV3 is our assumption that there exists an underlying hierarchical structure of achievements. We will explain our assumption and the difference between other baselines more clearly in the revised version.
>
> **Q3: I also found the log-scale on the axes of Figure 6 to be a bit confusing. Is a success rate < 0.01% really significant?**
>
> Thank you for your helpful comment. We report the individual success rates of achievements using a log-scale, following the practice in DreamerV3 [1]. A success rate of an achievement under 0.01% implies that this is unlocked only in a subset of individual runs, also mentioned in the second paragraph in Section 5.2 of our main paper. Nevertheless, collecting a diamond in an individual run remains noteworthy, as it is particularly challenging due to the scarcity of resources. We will explain this more clearly in the revised version.
>
> **Q4: Since crafter is partially observed, do you use a recurrent network?**
>
> In this paper, we do not employ recurrent neural networks such as LSTM and only utilize convolutional neural networks for image processing. LSTM has been widely utilized in RL to address partial observability [2, 3]. However, it has been noted that employing LSTM for PPO on Crafter does not yield a significant performance improvement, resulting in a score increase of only 0.1 [4]. Incorporating our contrastive learning with memory-based policies would be a promising avenue for future research.
>
> **Q5: Does PPO outperform DreamerV3 for the same number of environment steps? I would expect model-based to have higher sample efficiency, though it may have lower asymptotic performance.**
>
> We evaluate PPO and DreamerV3 with the same number of 1M environment steps and find that PPO outperforms DreamerV3. I agree with your opinion that model-based algorithms generally exhibit greater sample efficiency than model-free algorithms, provided the learned models are accurate.  However, in our attempt to reproduce the results of DreamerV3, we notice that the losses for training a world model (Equation 5 in the original paper) tend to increase as training progresses, leading to imprecise world models. Furthermore, we find a rapid drop in the policy entropy, reaching a level of 0.5 at timestep 50K and remaining low thereafter. This phenomenon hinders sufficient exploration for unlocking new achievements.
>
> ---
>
> We again thank the reviewer for providing constructive feedback, which truly enhances the quality of our paper. We hope that our response adequately addresses all the reviewer's questions.
>
> ---
>
> **References**
>
> [1] Danijar Hafner et al., Mastering Diverse Domains through World Models, arXiv 2023. \
> [2] Matthew Hausknecht and Peter Stone, Deep Recurrent Q-Learning for Partially Observable MDPs, arXiv 2015. \
> [3] Steven Kapturowski et al., Recurrent Experience Replay in Distributed Reinforcement Learning, ICLR 2019. \
> [4] Aleksandar Stanic et al., Learning to Generalize with Object-centric Agents in the Open World Survival Game Crafter, ToG 2023.

---

> > ### Comment · Reviewer_nAYF · 2023-08-10
> > **Thank you for your work!**
> >
> > I would like to thank the authors for continuing to improve their work. I find the new experiments across domains and with value-based algorithms convincing. I am consequently improving my score.
> >
> > I additionally think it would be of great value to the community if the authors could detail how they exactly implement the necessary achievement counting component of their method for new domains.

---

### Official Review · Reviewer_Tzgb · 2023-07-05

**Soundness:** 3 good
**Presentation:** 3 good
**Contribution:** 3 good
**Rating:** 7
**Confidence:** 3

**Summary:**

This work introduces achievement distillation, a model-free RL method designed to discover achievements without the need for explicit long-term planning components. The proposed method comprises three primary components: two self-supervised tasks and a memory component. The self-supervised tasks, namely Intra-trajectory achievement prediction and cross-trajectory achievement matching, guide the encoder to predict the next achievement to be unlocked using a contrastive learning loss function with different objectives. The memory component is formed by concatenating the latent state representation from the encoder with the action and the representation of the previous achievement. These components are integrated into the PPO algorithm through two alternating training phases involving policy training and auxiliary self-supervised tasks. The effectiveness of achievement distillation is evaluated in the Crafter environment, where it demonstrates significant performance improvements over strong baselines. Additionally, the authors conducted analyses on model sizes, representations, and the contribution of individual components.

**Strengths:**

1. The primary strength of the paper lies in the significant results it presents. The main results in Table 1 and Figure 5 provide clear evidence that the proposed achievement distillation method outperforms the baseline methods to a substantial degree. Furthermore, the results in Figure 6 demonstrate that achievement distillation achieves success in several achievements that none of the baselines can accomplish (e.g., making an iron sword), which is impressive.
2. The authors have done a great job in implementing various relevant strong baselines. Reproducing these baselines are non-trivial given the complexity of them.
3. While the concept of self-supervised auxiliary tasks and the used self-supervised losses are not novel ideas, the authors have managed to execute them very well, resulting in an agent with remarkable performance.
4. The paper is well-written, and the ideas are effectively presented and justified.


**Weaknesses:**

1. The primary weakness of this work lies in the limited range of environments in which the proposed method is evaluated. Since the authors only tested it on the Crafter environment, it remains uncertain whether their method would be effective in different settings and whether it has avoided overfitting to the Crafter environment.

2. The ablation study appears to be somewhat superficial. Although the contribution of cross-trajectory achievement matching is evident, the significance of cross-trajectory achieving matching and memory is not as convincing. The authors should provide additional evidence to support the role of these components. Please refer to the next section for suggestions on how to improve this aspect.

**Questions:**

1. I would encourage the authors to address the main limitation of their work discussed above by conducting further experimentation in various environments, such as MiniHack and maybe ProcGen.

2. It would greatly enhance the understanding of the contribution of cross-trajectory achieving matching and memory if the authors could conduct additional research. For instance, including the individual success rates for all achievements (Figure 6) while ablating these components could be a valuable option.

3. It would be interesting if the authors could present the results in Table 1 and Figure 5 for a larger range of environment steps (e.g., 5M, 10M). This would clarify whether this method is only more data efficient or remains superior with more steps.

4. In Figure 6, why the Individual success rate is not shown for MuZero+SPR?

5. Do you think your method would work with RL algorithms other than PPO?

**Limitations:**

The authors have adequately addressed the limitations.

---

> ### Author Rebuttal · Authors · 2023-08-09
>
> Thank you for your constructive feedback. We appreciate the encouraging comments (“The authors have done a great job in implementing various relevant strong baselines”, “The paper is well-written, and the ideas are effectively presented and justified”). We would like to address your questions below.
>
> ---
>
> **Q1: I would encourage the author to conduct further experimentation in various environments, such as MiniHack and maybe ProcGen.**
>
> Thank you for your suggestion. We conduct experiments on two additional benchmarks that feature hierarchical achievements: Procgen Heist and a custom MiniGrid environment. Please refer to the Global Response Q1 for more details on the benchmarks and our experimental results.
>
> **Q2: It would greatly enhance the understanding of the contribution of cross-trajectory achieving matching and memory if the authors could conduct additional research.**
>
> Thank you for the suggestion. We first provide the individual success rates for challenging achievements, such as collecting iron, while conducting ablation studies on cross-trajectory matching (C) and memory (M). The table below demonstrates that both cross-trajectory achievement matching and memory play significant roles in unlocking these challenging achievements.
>
> |Achievement|PPO + I|PPO+I+C|PPO+I+C+M|
> |---|---|---|---|
> |Make stone pickaxe|10.92|16.43|22.93|
> |Make stone sword|14.32|20.08|23.35|
> |Collecting iron|1.33|2.70|4.02|
> |Make iron pickaxe|0.01|0.00|0.01|
> |Make iron sword|0.00|0.00|0.02|
>
> However, we find that there is no substantial difference in the success rates for easy achievements, such as crafting wooden tools, as shown below.
>
> |Achievement|PPO + I|PPO+I+C|PPO+I+C+M|
> |---|---|---|---|
> |Make wood pickaxe|71.42|71.44|72.69|
> |Make wood sword|67.16|68.68|70.86|
>
> Additionally, to gain deeper insight into how the cross-trajectory matching works, we conduct an oracle experiment suggested by Reviewer cpyF. In this experiment, we replace the cross-trajectory Wasserstein matching with exact matching using oracle achievement labels and compare its performance with our matching algorithm. Table 2 in the attached file presents the performance of oracle cross-trajectory matching. From this analysis, we anticipate a score increase of 1.8 when the cross-trajectory matching is optimized to its fullest potential.
>
> **Q3: It would be interesting if the authors could present the results in Table 1 and Figure 5 for a larger range of environment steps.**
>
> Thank you for your valuable recommendation. We increase the number of environment steps to 10M and evaluate the performance of our method against three baseline: PPO, LSTM-SPCNN, and SEA. It is worth noting that training a DreamerV3 agent with 10M environment steps was not feasible within the constrained time of the rebuttal period. With a single NVIDIA RTX 3090 GPU, it requires approximately 2.5 days to complete just 1M environment steps. As illustrated in Figure 6 of the attached file, our method not only demonstrates higher sample efficiency but also outperforms the baselines with superior final performance.
>
> **Q4: In Figure 6, why the Individual success rate is not shown for MuZero + SPR?**
>
> We first clarify that the results of MuZero + SPR in this paper have been derived from the original paper due to the unavailability of its source code. Additionally, the original paper only presents the individual success rate for a MuZero + SPR agent with pre-training using 150M exploratory data. Since our primary focus is on training an agent without pre-training, we have opted not to include MuZero + SPR in Figure 6. Afterwards, we plan to reproduce the results of MuZero + SPR and provide its individual success rate once the official source code becomes available.
>
> **Q5: Do you think your method would work with RL algorithms other than PPO?**
>
> Thank you for bringing this to our attention. We evaluate our contrastive learning method alongside a popular off-policy value-based algorithm QR-DQN and observe its strong performance. For more details on the experimental settings and results, please refer to General Response 2.
>
> ---
>
> Once again, we really appreciate the reviewer's insightful questions, which greatly help us to to enhance our paper. We hope that our response above addresses all of the reviewer's questions.

---

> > ### Comment · Reviewer_Tzgb · 2023-08-13
> >
> > I would like to express my gratitude to the authors for investing the time in conducting these additional experiments. These efforts serve to underscore the importance of their work and also lead me to elevate my rating accordingly.

---

### Official Review · Reviewer_z85w · 2023-07-07

**Soundness:** 3 good
**Presentation:** 3 good
**Contribution:** 2 fair
**Rating:** 6
**Confidence:** 4

**Summary:**

This paper proposes a contrastive learning approach for representation learning in the problem of hierarchical achievement discovery. The proposed method leverages previous contrastive learning loss and combines it with optimal transport. Empirical results show that the learned representation could improve PPO in the Crafter environment.

**Strengths:**

1.	It is interesting that self-supervised representation learning can improve PPO, which significantly outperforms model-based approaches, even regarding sample efficiency.
2.	Experimental results show the strong performance of the proposed method in the Crafter environment.
3.	Presentation: this paper is generally well-organized and easy to follow.

**Weaknesses:**

1.	Domain knowledge: unlike baselines, the proposed method utilizes additional important knowledge, i.e., identifying when a new achievement is unlocked (which can be easily inferred from the observed rewards). This other knowledge can easily explain why the achievement prediction of the proposed method is much better than PPO.
2.	Generalization: this paper only shows its results in the Crafter environment. It is highly recommended to conduct experiments in other environments to show its generality.
3.	Novelty: though it is interesting to use optimal transport for matching achievements, the proposed method is a natural application of contrastive learning, which is not quite novel.

**Questions:**

1.	This paper defines MDPs with hierarchical achievements. Has prior work studied this model? If so, please add references.
2.	It is interesting to see PPO is more sample-efficient than model-based methods in this paper’s experiments. Except for implementation practice improvement of PPO, any other insights for this result?

**Limitations:**

This paper has discussed its limitation.

---

> ### Author Rebuttal · Authors · 2023-08-09
>
> We appreciate the review's constructive and helpful feedback. We are encouraged by the reviewer’s positive comments (“it is interesting that self-supervised representation learning can improve PPO”, “this paper is generally well-organized and easy to follow”). We would like to address the questions and concerns raised by the reviewer, as detailed below.
>
> ---
>
> **Q1: The proposed method utilizes additional important knowledge.**
>
> We agree with your view that our method utilizes additional information about the reward structure, where each reward signal represents distinct achievements. Nonetheless, we would like to emphasize that we employ the same reward function as the baseline methods without any modification and do not leverage any other information regarding achievements beyond the reward function itself.
>
> **Q2: It is highly recommended to conduct experiments in other environments to show its generality.**
>
> Thank you for your recommendation. We conduct experiments on two additional benchmarks that feature hierarchical achievements: Procgen Heist and a custom MiniGrid environment. Please refer to the Global Response Q1 for more details on the benchmarks and our experimental results.
>
> **Q3: This paper defines MDPs with hierarchical achievements. Has prior work studied this model? If so, please add references.**
>
> The concept of MDPs with hierarchical achievements has been studied in recent prior work [1, 2]. While we have cited these studies in the related work section, we will also include these references in Section 2.1 for further clarification. Thank you for the suggestion.
>
> **Q4: It is interesting to see PPO is more sample-efficient than model-based methods in this paper’s experiments. Except for implementation practice improvement of PPO, any other insights for this result?**
>
> In general, model-based algorithms exhibit greater sample efficiency than their model-free counterparts, provided that the learned models are accurate. However, training accurate world models on procedurally generated, partially observable environments poses a significant challenge. When reproducing the results of DreamerV3, we find that the losses for the world model gradually increase as training continues. It is worth noting that an agent in Crafter encounters a new environment in each episode. Given this, the world model of DreamerV3, which relies heavily on prior experience from the replay buffer for training, does not transfer well to unseen environments. This challenge has been underscored by a recent study, which further demonstrates that a Dreamer agent struggles to adapt rapidly to the ever-changing environments [3]. In contrast, PPO usually updates the policy and value networks using Monte Carlo methods with recently collected episodes, and does not heavily rely on models trained with prior experience, compared to DreamerV3.
>
> Additionally, we observe a rapid drop in the policy entropy of DreamerV3, reaching a level of 0.5 at timestep 50K and remaining low thereafter. In contrast, PPO maintains the policy entropy above a level of 1.0 by the end of the training process. This phenomenon limits a DreamerV3 agent's ability to explore environments and unlock new achievements.
>
> ---
>
> We again thank the reviewer for giving constructive suggestions. We hope our explanation above addressed all reviewer's questions.
>
> ---
>
> **References**
>
> [1] Robby Costales et al., Possibility Before Utility: Learning And Using Hierarchical Affordances, ICLR 2022. \
> [2] Zihan Zhou and Animesh Garg, Learning Achievement Structure for Structured Exploration in Domains with Sparse Reward, ICLR 2023. \
> [3] Isaac Kauvar et al., Curious Replay for Model-based Adaptation, ICML 2023.

---

> > ### Comment · Reviewer_z85w · 2023-08-18
> >
> > I appreciate the authors' detailed response, which partially addresses my concerns. However, I am not entirely convinced that PPO would be more sample-efficient than Dreamer, as Dreamer seemed able to learn good models in the experimental environment. I guess it is possible that DreamerV3 has much more parameters than the proposed methods. I wonder if the authors have tried fewer parameters for DreamerV3. Anyway, I will raise my score to 6.

---

### Author Rebuttal · Authors · 2023-08-09

We thank all the reviewers for their time and effort in providing valuable feedback. We are especially appreciative of the encouraging comments we received from each of them. To begin our response, we would like to first address some of the common concerns that have been raised by multiple reviewers.

---

**Q1: Application to other environments**

We conduct experiments on two additional benchmarks featuring hierarchical achievements: Procgen Heist [1] and a custom MiniGrid environment [2].

Heist is a procedurally generated “Door Key” environment where the goal is to steal a gem hidden behind a sequence of blue, green, and red locks, as illustrated in Figure 1 of the attached file. To open each lock, an agent must collect a key with the corresponding color. We consider opening each lock and stealing a gem as achievements. It is worth noting that Heist introduces another challenge, given that the color of wall and background can vary between environments, whereas Crafter maintains fixed color patterns for its terrains. Moreover, there is only a single pathway to unlock an achievement in Heist, while multiple routes exist to unlock an achievement in Crafter. For instance, an agent can readily gather wood almost everywhere on the map due to its abundance. To ensure closer alignment with Crafter, we slightly adjust the reward structure so that an agent receives a reward of 2 for opening each lock and a reward of 10 for successfully stealing a gem. We train an agent in the “hard” difficulty mode for 25M environment steps and evaluate its performance in terms of the success rate of gem pilfering and the episode reward.

Additionally, we create a customized "Door Key" environment using the MiniGrid library to assess the effectiveness of our method on a larger achievement graph. The design of this environment takes inspiration by TreeMaze proposed in SEA [3]. An agent must sequentially unlock doors, find keys, and ultimately reach the green square, as depicted in Figure 2 of the attached file. The environment comprises a total of 10 achievements, such as opening doors, collecting keys, and reaching the goal. An agent receives a reward of 1 for unlocking a new achievement, mirroring the reward structure in Crafter. We train an agent for 1M environment steps and evaluate its performance in terms of the geometric mean of success rates and the episode reward, following the same protocol as Crafter.

Figure 3 and 4 of the attached file illustrate the performance of our method and PPO. Remarkably, our method significantly enhances the performance of PPO in Heist, elevating the success rate from 29.6% to 71.0%. Furthermore, our method outperforms PPO in the MiniGrid environment by a substantial margin, increasing the score from 3.33% to 8.04%. These results highlight the broad applicability of our method to diverse environments with hierarchical achievements.

**Q2: Application to value-based RL algorithms**

We evaluate our contrastive learning method in conjunction with a popular off-policy value-based algorithm QR-DQN [4] on Crafter for 1M environment steps and observe its strong performance. Specifically, we apply our contrastive learning to the Q-network encoder at intervals of every 8000 environment steps. We employ Huber quantile regression to preserve the Q-network’s output distribution in a manner congruent with the value function optimization in QR-DQN. Figure 5 of the attached file demonstrates that our contrastive learning method is also effective in value-based algorithms, enhancing the performance of QR-DQN from 4.14 to 8.07.

---

We hope that our response addresses all the questions and concerns of the reviewers. Please let us know if there are any further questions.

---

**References**

[1] Karl Cobbe et al., Leveraging Procedural Generation to Benchmark Reinforcement Learning, ICML 2020. \
[2] Maxime Chevalier-Boisvert et al., Minigrid & Miniworld: Modular & Customizable Reinforcement Learning Environments for Goal-Oriented Tasks, arXiv 2023. \
[3] Zihan Zhou and Animesh Garg, Learning Achievement Structure for Structured Exploration in Domains with Sparse Reward, ICLR 2023. \
[4] Will Dabney et al., Distributional Reinforcement Learning with Quantile Regression, AAAI 2018.

---

### Decision · Program_Chairs · 2023-09-21

**Decision:**

Accept (poster)

**Comment:**

This is a nice paper, and the reviews were all positive. I recommend accepting it.